# High-Throughput Drug Library Screening in Primary *KMT2A*-Rearranged Infant ALL Cells Favors the Identification of Drug Candidates That Activate P53 Signaling

**DOI:** 10.3390/biomedicines10030638

**Published:** 2022-03-10

**Authors:** Priscilla Wander, Susan T. C. J. M. Arentsen-Peters, Kirsten S. Vrenken, Sandra Mimoso Pinhanҫos, Bianca Koopmans, M. Emmy M. Dolman, Luke Jones, Patricia Garrido Castro, Pauline Schneider, Mark Kerstjens, Jan J. Molenaar, Rob Pieters, Christian Michel Zwaan, Ronald W. Stam

**Affiliations:** 1Princess Máxima Center for Pediatric Oncology, 3584 CS Utrecht, The Netherlands; p.wander@erasmusmc.nl (P.W.); t.c.j.peters-3@prinsesmaximacentrum.nl (S.T.C.J.M.A.-P.); k.s.vrenken@prinsesmaximacentrum.nl (K.S.V.); spinhancos@cnc.uc.pt (S.M.P.); b.koopmans@prinsesmaximacentrum.nl (B.K.); edolman@ccia.org.au (M.E.M.D.); lukej4643@gmail.com (L.J.); p.garridocastro@gmail.com (P.G.C.); p.schneider@prinsesmaximacentrum.nl (P.S.); j.j.molenaar@prinsesmaximacentrum.nl (J.J.M.); r.pieters@prinsesmaximacentrum.nl (R.P.); c.m.zwaan@prinsesmaximacentrum.nl (C.M.Z.); 2Department of Pediatric Oncology/Hematology, Erasmus MC-Sophia Children’s Hospital, 3015 CN Rotterdam, The Netherlands; markkerstjens@gmail.com; 3CNC-Center for Neurosciences and Cell Biology, University of Coimbra, 3004-504 Coimbra, Portugal; 4Children’s Cancer Institute, Lowy Cancer Centre, UNSW Sydney, Kensington, Sydney, NSW 2052, Australia; 5School of Women’s and Children’s Health, Faculty of Medicine, University of New South Wales, Sydney, NSW 2031, Australia; 6Department of Pharmaceutical Sciences, Utrecht University, 3584 CS Utrecht, The Netherlands

**Keywords:** MLL-rearrangements, acute lymphoblastic leukemia, infant ALL, p53, drug library screening

## Abstract

*KMT2A*-rearranged acute lymphoblastic leukemia (ALL) in infants (<1 year of age) represents an aggressive type of childhood leukemia characterized by a poor clinical outcome with a survival chance of <50%. Implementing novel therapeutic approaches for these patients is a slow-paced and costly process. Here, we utilized a drug-repurposing strategy to identify potent drugs that could expeditiously be translated into clinical applications. We performed high-throughput screens of various drug libraries, comprising 4191 different (mostly FDA-approved) compounds in primary *KMT2A*-rearranged infant ALL patient samples (*n* = 2). The most effective drugs were then tested on non-leukemic whole bone marrow samples (*n* = 2) to select drugs with a favorable therapeutic index for bone marrow toxicity. The identified agents frequently belonged to several recurrent drug classes, including BCL-2, histone deacetylase, topoisomerase, microtubule, and MDM2/p53 inhibitors, as well as cardiac glycosides and corticosteroids. The in vitro efficacy of these drug classes was successfully validated in additional primary *KMT2A*-rearranged infant ALL samples (*n* = 7) and *KMT2A*-rearranged ALL cell line models (*n* = 5). Based on literature studies, most of the identified drugs remarkably appeared to lead to activation of p53 signaling. In line with this notion, subsequent experiments showed that forced expression of wild-type p53 in *KMT2A*-rearranged ALL cells rapidly led to apoptosis induction. We conclude that *KMT2A*-rearranged infant ALL cells are vulnerable to p53 activation, and that drug-induced p53 activation may represent an essential condition for successful treatment results. Moreover, the present study provides an attractive collection of approved drugs that are highly effective against *KMT2A*-rearranged infant ALL cells while showing far less toxicity towards non-leukemic bone marrow, urging further (pre)clinical testing.

## 1. Introduction

Acute lymphoblastic leukemia (ALL) is the most common type of childhood cancer for which the prospects have improved substantially over the past decades, with the overall survival rates increasing from <10% in the 1960s to >90% today [1]. Despite these advances, infants (i.e., children <1 year of age) diagnosed with ALL fare significantly worse. In ~80% of the cases, infant ALL is driven by chromosomal translocations involving the *KMT2A* (formerly known as *MLL*) gene, which are associated with a poor clinical outcome. Consequently, the 5-year event-free survival (EFS) rates for *KMT2A*-rearranged infant ALL persistently remained dismal at 30 to 40% over the last two decades [2,3].

High-throughput genomics and profiling studies have significantly contributed to uncovering the unique biology of *KMT2A*-rearranged infant ALL, defining its distinctive gene expression and DNA methylation profiles [4,5,6,7,8,9] and elucidating its remarkably silent mutational landscape [10,11]. Yet, the number of suitable therapeutic targets identified by such studies remained limited and, despite substantial preclinical evaluations, rarely led to meaningful clinical trials. A notable exception would be targeting of the histone methyltransferase DOT1L, the key effector of KMT2A fusion-mediated leukemic transformation [12]. Unfortunately, the response rates to the DOT1L inhibitor Pinometostat (EZP-5676) in an adult phase I trial (NCT02141828) were rather modest and disappointing [13]. In general, drug development in cancer has proven to be difficult, with only a 5% likelihood of approval and an average time span of 11.4 to 13.5 years [14,15]. Hence, while promising clinical trials testing the efficacy of Menin inhibition [16] (NCT04067336) or immunotherapies, e.g., Blinatumomab (NTR6359) and CAR-T cells [17], against *KMT2A*-rearranged infant ALL are being initiated, readily available and effective therapeutic alternatives to standard combination chemotherapy hardly exist.

Drug repurposing has become a popular strategy in drug discovery as extensive data on toxicity and pharmacokinetics are already available. This reduces the laborious preclinical studies as well as the risk of failure in early clinical trials [18], although drugs are not always developed for this specific population, or no suitable pediatric formulation may be available. Various examples of successful drug repurposing have been summarized by Pushpakom et al. [19]. For instance, Raloxifene, originally indicated to treat and prevent osteoporosis, readily received FDA approval for preventing invasive breast cancer [20]. Given the potential benefits of drug repurposing and the urgent need for more adequate treatment options for *KMT2A*-rearranged infant ALL, we recently performed an extensive drug library screen on various leukemia cell line models [21]. That study revealed that Irinotecan exhibited strong anti-leukemic effects against *KMT2A*-rearranged ALL in vitro as well is in vivo, inducing complete remissions in various mouse models. Encouraged by these results, we decided here to additionally perform drug library screens on primary *KMT2A*-rearranged infant ALL samples. This led to the identification of several interesting drug candidates. Interestingly, we found that the majority of the drugs were effective against *KMT2A*-rearranged infant ALL while largely sparing non-leukemic bone marrow cells, which directly or indirectly affect p53 signaling. Apart from providing numerous attractive drug candidates for further preclinical testing, our data suggest that *KMT2A*-rearranged ALL may well be vulnerable to the activation of p53.

## 2. Materials and Methods

### 2.1. Patient Samples

Primary *KMT2A*-rearranged infant ALL patient samples were collected as part of the INTERFANT treatment studies [2,3]. Informed consent was obtained from parents or legal guardians to use excess diagnostic material for research purposes, as approved by the institutional review board. One of the non-leukemic bone marrow (BM) aspirates was from a pediatric patient suspected to suffer from T-cell lymphoma but who did not show any signs of infiltration in the BM. The second non-leukemic BM aspirate was from a pediatric ALL patient in complete remission. All samples were processed and cultured as described elsewhere [22]. The patient-derived leukemic samples used in this study all contained >90% blasts, and both non-leukemic BM controls contained <1% blasts, as determined by May-Grünwald Giemsa (Merck, Darmstadt, Germany) stained cytospins.

### 2.2. Cell Line Cultures

The *KMT2A*-rearranged ALL cell lines RS4;11, ALL-PO, BEL-1, and SEM all carry the t(4;11) translocation, giving rise to *KMT2A–AFF1* fusion transcripts. *KMT2A*-rearranged ALL cell line KOPN-8 expresses the *KMT2A-MLLT1* fusion gene as a consequence of a t(11;19) translocation. RS4;11 was purchased from ATCC (Manassas, VA, USA), SEM, KOPN-8, and HEK293T were purchased from DSMZ (Braunschweig, Germany), ALL-PO was a gift from Dr. Cazzaniga (University of Milano-Bicocca, Milano MI, Italy), and BEL-1 was a gift from Dr. Ruoping Tang (University Laboratory, Paris, France). RS4;11, ALL-PO, BEL-1, SEM, and KOPN-8 were maintained in RPMI-1640 with GlutaMAX, and HEK293T cells was maintained in Dulbecco’s modified Eagle medium. All media were supplemented with 10% fetal calf serum, 100 IU/mL penicillin, 100 µg/mL streptomycin, and 0.125 µg/mL Amphotericin (Life Technologies, Carlsbad, CA, USA) and cultured at 37 °C in humidified air containing 5% CO2, except for BEL-1, which required 20% fetal calf serum instead of 10%. Mycoplasma testing and DNA fingerprinting were regularly performed to ensure the quality and integrity of our cell lines.

### 2.3. High-Throughput Drug Library Screening and Viability Assays

Cells were semi-automatically seeded in 384-well plates (Corning, NY, USA) using a MultidropTM dispenser (Thermo Fisher Scientific, Waltham, MA, USA). For high-throughput drug screening, drugs were added to the 384-well plates using a SciClone ALH3000 liquid handling robot (Caliper Life Sciences, Waltham, MA, USA) to a final concentration of 10 nm, 100 nm, or 1000 nm. All tested drugs came from commercially available drug libraries, including the Prestwick Chemical library (Prestwick Chemical Libraries, Illkirch, France), the anti-neoplastic sequoia library (Sequoia Research Products, Pangbourne, UK), the Epigenetics library (Enzo Life Sciences, Zandhoven, Belgium), the Epigenetics screening library (Cayman Chemical, Ann Arbor, MI, USA), the Spectrum collection (Microsource, Gaylordsville, CT, USA), and the Cell cycle/DNA Damage compound library (MedChemExpress, Princeton, NJ, USA). Apart from these drug libraries, several drugs were selected for further validation and additional experiments, including HC toxin (Cayman Chemical, Ann Arbor, MI, USA), Idasanutlin (Roche, Basel, Switzerland), AMG232 and Milademetan (MedChemExpress, Princeton, NJ, USA), and SN-38, YM155, UNC0321, SP2509, Isoliquiritigenin, Neratinib, Venetoclax, and Navitoclax (Selleckchem, Houston, TX, USA). For validation experiments, broad range dose–response curves were generated using a Tecan D300 Digital Dispenser (Tecan, Männedorf, Switzerland). DMSO dissolved drug stocks were diluted in non-supplemented RPMI, allowing a maximum DMSO concentration of ≤0.5% (*v/v*). Cell viability was assessed by 4-day thiazolyl blue tetrazolium bromide (MTT; Sigma-Aldrich, Saint Louis, MO, USA) assays, as described elsewhere [23]. Drug response data were analyzed using GraphPad Prism 8 (GraphPad Software, San Diego, CA, USA). Data were normalized against vehicle controls, i.e., cells cultured in absence of drugs.

### 2.4. Western Blotting

Whole cell protein lysates were resolved on precast SDS-polyacrylamide gels (TGX, Bio-Rad, Hercules, CA, USA) and separated proteins were transferred to nitrocellulose membranes using a Transblot Turbo Transfer System (Bio-Rad, Hercules, CA, USA). Membranes were blocked in 5% milk (Elk, Campina, Zaltbommel, The Netherlands) and subsequently probed with mouse or rabbit antibodies directed against p53, acetylated p53 (at K382), p21, MDM2, cleaved PARP, and GAPDH (Cell Signaling, Danvers, MA, USA), followed by IRDye 680RD/800CW conjugated secondary antibodies (LI-COR Biosciences GmbH, Bad Homburg vor der Höhe, Germany). Membranes were scanned on the LI-COR Odyssey imaging system. Western blots were quantified by analyzing the intensity of the western blot band compared to loading control using Image Studio Lite software.

### 2.5. Cloning of the P53 Wild-Type and P53 Mutant into an Inducible Vector

Plasmids pMSCV-YFP-P53-WT and pMSCV-YFP-P53-R248 were a kind gift from the lab of F.N. van Leeuwen (Princess Máxima Center, Utrecht, The Netherlands). FLAG-P53-WT and FLAG-P53-R248Q mutant cDNA with flanking restriction sites for NheI and BamHI were produced from these plasmids by PCR. To produce inducible wild-type p53 and mutant (R248Q) p53 expressing lentiviral plasmids, the cDNA products were cloned into the pCW-Cas9 vector (#50661, Addgene, Watertown, MA, USA) by replacing Cas9 with the FLAG-P53-WT cDNA products using NheI (New England Biolabs, Ipswich, MA, USA) and BamHI (Promega, Madison, WI, USA) restriction enzymes. The produced pCW-FLAG-P53-WT and pCW-FLAG-P53-R248Q vectors were verified by sequencing.

### 2.6. Transfection, Lentivirus Production, and Transduction

To generate inducible p53 wild-type and p53 mutant *KMT2A*-rearranged ALL cell lines, vectors pCW-FLAG-P53-WT or pCW-FLAG-P53-R248Q were used. These vectors were transfected in HEK293T cells using 1.9 µg pMD2.G, 3.7 µg psPAX2 (Addgene, Watertown, MA, USA), and 55.5 µg PEI 25K™ (Polysciences, Warrington, PA, USA) in Opti-MEM. Virus was harvested 48 h after transfection and concentrated using the vivaspin-20 columns (Sigma-Aldrich, Saint Louis, MO, USA). Concentrated virus was stored at −80 °C until further use. Thereafter, FLAG-P53-WT and FLAG-P53-R248Q transgenes were transduced into RS4;11 and BEL-1 cells. The p53 BEL-1 cell lines were transduced by spin transduction, whereas spin transduction of RS4;11 cells was aided by precoating the plates with retronectin (Takara Bio Europe, Saint-Germain-en-Laye, France). Cells were kept in their normal culture medium containing Tetracycline free FCS prior to, during, and after transduction. Cells were selected using 1 µg/mL puromycin (Sigma-Aldrich, Saint Louis, MO, USA). Expression of the p53 wild-type and p53 mutant transgenes were induced by addition of 1 and 2 µg/mL doxycycline (Sigma-Aldrich, Saint Louis, MO, USA) to the culture medium of transduced BEL-1 and RS4;11 cells, respectively. Cells were counted manually using Trypan Blue exclusion and harvested for western blot analyses at indicated timepoints. For apoptosis measurements, cells were incubated for 24 h with doxycycline before flow cytometry analyses.

### 2.7. Apoptosis Assay

Cells were stained with PE Annexin V and 7AAD using the PE Annexin V Apoptosis Detection I Kit (BD Pharmingen, San Diego, CA, USA) according to manufacturer’s protocol. Apoptotic cells were detected by measuring the amount of Annexin V and 7AAD positive cells using the flow cytometer CytoFLEX LX (Beckman Coulter, Woerden, The Netherlands) followed by data analyses using the CytExpert software (Beckman Coulter, Woerden, The Netherlands).

## 3. Results

### 3.1. Drug Library Screens on Primary KMT2A-Rearranged Infant ALL Cells Reveal Attractive Candidate Drugs Favoring P53 Activation

Various drug libraries, comprising a total of 4191 different compounds, were used to screen for drugs effective against two primary *KMT2A*-rearranged infant ALL samples in vitro at 10 nm, 100 nm, and 1000 nm drug concentrations. One of the patient samples was characterized by a KMT2A–AFF1 fusion protein, the most common *KMT2A* translocation among infant ALL patients [24], and the other sample carried a KMT2A–MLLT11 fusion (as a result of a t(1;11)(q21;q23) translocation). Individual drugs were defined as ‘hits’ when the cell viability in both patient samples was less than 30%, 40%, and 50% at drug concentrations of 1000 nm, 100 nm, and 10 nm, respectively. Thereafter, identified ‘hits’ were tested on two non-leukemic whole pediatric bone marrow (BM) samples to allow for the selection of drug candidates with a favorable therapeutic index for bone marrow toxicity. Drug ‘hits’ showing a cell viability of >60% in both non-leukemic BM samples were considered as potential drug candidates (Figure 1A). Among the identified drug candidates, the most recurrent types of drugs (or drug classes) included: BCL-2 inhibitors, histone deacetylase (HDAC) inhibitors, topoisomerase inhibitors, cardiac glycosides, MDM2/p53 inhibitors, microtubule inhibitors, and corticosteroids (Figure 1B). At the 10 nm drug concentration, we identified Venetoclax, a well-studied BCL2 inhibitor known to be highly active against *KMT2A*-rearranged acute leukemia [25,26]. Moreover, HDAC inhibition also has been shown to be highly effective against *KMT2A*-rearranged ALL [27,28,29]. Taken together, this strongly supports the validity of our drug screen.

Apart from identifying agents known to be effective against *KMT2A*-rearranged acute leukemia, our drug screen also provides numerous novel candidate drugs (Figure 1A). As these agents were selected to be highly active against *KMT2A*-rearranged infant ALL cells while largely sparing non-leukemic BM cells, this screen represents a heterogeneous collection of attractive drug candidates for further exploration in subsequent preclinical studies. Moreover, investigating the mechanism of action of the candidate drugs described in the literature showed that the majority of the here-identified drug candidates appear to inhibit proteins that normally repress p53 activation. In other words, most of the identified drugs all seem to (in)directly lead to the activation of p53 signaling (Figure 1C). For instance, HDAC inhibitors can induce accumulation of p53 and suppress MDM2 expression, a p53 interacting protein that directly targets p53 for proteasomal degradation, and inhibit transcriptional activity of p53 [30]. Likewise, SN-38 (the active metabolite of Irinotecan) is able to induce p53 expression and p53 phosphorylation [31], and Survivin inhibition has been shown to increase p53-dependent apoptosis [32]. G9a is able to methylate p53 at Lys(373), which correlates with inactive p53 [33], and Lysine-specific histone demethylase 1A (LSD1) has been shown to interact with p53 to repress p53-mediated transcriptional activation [34]. Isoliquiritigenin was shown to induce apoptosis through upregulating p53 expression [35], and Idasanutlin directly abrogates the MDM2/p53 interaction, which leads to p53 stabilization and activity [36].

### 3.2. Validation of Candidate Drugs on Additional Primary KMT2A-Rearranged Infant ALL Samples and KMT2A-Rearranged ALL Cell Lines

To further emphasize the validity of our drug screen, we generated full dose–response curves for several drugs covering most of the identified drug classes (Figure 1B), including HC toxin (an HDAC inhibitor) [37], Idasanutlin (an MDM2 inhibitor) [38], SN-38 (the active metabolite of Irinotecan) [39], YM155 (a Survivin inhibitor) [40], UNC0321 (a G9a histone methyltransferase inhibitor) [41], SP2509 (an LSD1 inhibitor) [42], Isoliquiritigenin (an aldose reductase and FLT3 inhibitor) [43], and Neratinib (a pan-HER2 inhibitor) [44]. For this, 4-day MTT-assays were performed using seven additional primary *KMT2A*-rearranged infant ALL samples (Figure 2A), as well as five *KMT2A*-rearranged ALL cell line models (Figure 2B). For most of these selected drugs, the drug responses were comparable to those of the samples used in the initial drug library screen, except for Isoliquiritigenin and UNC0321. Both the additional primary *KMT2A*-rearranged infant ALL samples as well as the selected cell line models only modestly responded to Isoliquiritigenin and completely failed to respond to UNC0321 (Figure 2A,B). This is probably related to technical aberrations of the high-throughput screening, with UNC0321 being a false positive hit. Furthermore, among the *KMT2A*-rearranged ALL cell lines, both SEM and ALL-PO did not respond to Idasanutlin. This can be explained by the fact that Idasanutlin is a direct inhibitor of MDM2 and that both SEM and ALL-PO carry *p53* mutations, resulting in conformational changes, the loss of its tumor suppressor functions, and conveying oncogenic gain-of-function activities. Both cell lines also appeared to be slightly less sensitive to SN-38 and YM155. This, and the fact that the presence of a *p53* mutation is often associated with increased resistance to chemotherapy [45], may suggest that the here-identified drug candidates may be less suitable for patients carrying mutated *p53*. However, among infants diagnosed with *KMT2A*-rearranged ALL, the presence of *p53* mutations is rare: Agraz-Doblas et al. [10] identified *p53* mutations in 2 out of 49 patient samples, and Andersson et al. [11], found 2 out of 47 primary *KMT2A*-rearranged infant ALL samples to carry a mutation in *p53*. All of the patient samples used in this study carried wild-type *p53* (data not shown).

### 3.3. Selected Drug Candidates Affect P53 Signaling in KMT2A-Rearranged ALL Cells

As the anti-tumor effects of most of the here-identified drug candidates involve activation of p53, we determined the effects on p53 signaling upon short-term exposures to selected drugs in the *KMT2A*-rearranged ALL cell lines RS4;11 and ALL-PO. RS4;11 carries wild-type *p53*, whereas ALL-PO carries a missense mutation (R248Q) in exon 7 of the *p53* gene (Appendix A). Both cell lines were exposed to high concentrations of HC toxin, Idasanutlin, SN-38, YM155, UNC0321, SP2509, Isoliquiritigenin, and Neratinib to evaluate early effects on p53 signaling. For this, western blot analysis was used to investigate the influences on the expression of MDM2, p53 and p53 acetylation (lysine 382), p21, and PARP cleavage. Because we could not verify the drug responses to UNC0321 (Figure 2) indicating that this agent likely represents a false positive in our drug library screen, we decided to further include UNC0321 in our experiments as a negative control. In RS4;11, p53 expression is barely visible under normal conditions. When exposed to the drug candidates, all the drugs induced PARP cleavage (indicating apoptosis induction) and changes in expression of at least one of the proteins involved in p53 signaling as compared to vehicle controls (Figure 3A). Apart from Idasanutlin, all the drugs led to decreased expression of MDM2, the negative regulator of p53. At 6 h of exposure, Idasanutlin appears to induce an increase in MDM2 expression; however, additional experiments with extended time courses showed that MDM2 expression, as well as p53 acetylation and downstream target p21, increased at 3 h post exposure, and, therefore, probably activated the negative feedback mechanism mediated by p53 (Appendix A). For Idasanutlin, SN-38, Isoliquiritigenin, and, slightly for YM155, p53 expression became detectable accompanied by p53 acetylation at lysine 382, generally indicating an open conformation of p53, which enhances p53-mediated transcriptional activity [46]. SP2509 exposure did not show p53 expression but did induce detectable levels of acetylated p53 (Figure 3A). With the exception of SP2509, all the drugs induced an upregulation in p21 expression, indicating that p53-mediated transcriptional activity took place [47], in addition to apoptosis induction.

As expected, ALL-PO cells already displayed p53 expression and p53 acetylation prior to drug exposure (Figure 3B). The R248Q mutation in p53, as detected in ALL-PO, inhibits the wild-type p53 function by abrogating most of the cellular responses mediated by wild-type p53, and the R248Q mutation has a gain-of-function activity, i.e., the ability to transactivate novel cell-survival genes [48]. Yet, most drugs were able to inhibit MDM2 expression and induce elevated levels of p53 acetylation (Figure 3B). Induction of p21 expression was only observed in response to HC Toxin, suggesting ALL-PO cells respond differently to the drugs than RS4;11 cells. Idasanutlin and negative control UNC0321 had no effect on p53 signaling in ALL-PO, which is in concordance with the dose–response curves for Idasanutlin and UNC0321 to which ALL-PO cells are completely resistant (Figure 2B). Apart from Idasanutlin and UNC0321, SP2509 failed to induce PARP cleavage, which otherwise was observed for all the remaining agents (Figure 3B).

Taken together, except for Idasanutlin, all the selected candidate drugs affected the p53 pathway regardless of the R248Q mutation in p53, which ALL-PO carries. The varying effects on p53 signaling largely depended on the drug and may additionally be dependent on the chosen drug concentrations and selected time points of evaluation.

### 3.4. Induction of P53 Leads to Apoptosis of KMT2A-Rearranged ALL Cells

To further investigate the activation of p53 as a vulnerability signal to *KMT2A*-rearranged ALL cells, we transduced the BEL-1 cell line (carrying wild-type p53) with doxycycline-inducible expression vectors either encoding wild-type or mutated (R248Q) p53. Next, the effects of doxycycline-induced *p53* mRNA expression on MDM2, p21, and p53 protein expression and acetylation as well as PARP cleavage were determined. The induction of wild-type *p53* resulted in an increase in p53 protein expression, p53 acetylation, as well as activation of p21, MDM2, and substantial PARP cleavage, already at 6 h post induction (Figure 4A and Appendix A). At 16 h and 24 h, the p53 expression and acetylation decreased in the wild-type *p53* condition, probably because of a negative feedback mechanism. During the negative feedback mechanism, p53 stimulates the expression of MDM2, which is observed in BEL-1 at 6 h post induction. Subsequently, MDM2 ensures a ubiquitin-mediated proteasomal degradation of p53, and blocks the transcriptional activity of p53, causing the decreased expression in p53 and p53 acetylation observed in BEL-1 at 16 and 24 h post induction [49]. The induction of mutant *p53* led to p53 protein expression and acetylation but had no effect on MDM2 or p21 expression (Figure 4A). In addition, we determined the percentage of viable and apoptotic cells using flow cytometry 24 h post doxycycline induction of wild-type *p53* expression. In line with the observed PARP cleavage, this showed a ~35% decrease in viable BEL-1 cells and a ~35% increase in apoptosis (Figure 4B). Trypan blue exclusions staining confirmed these results (Figure 4C,D). We initially chose to conduct these experiments in BEL-1 cells as this cell line can rather effectively be transduced as compared to, for instance, RS4;11. Nonetheless, we also performed similar experiments using RS4;11, albeit with low transduction efficiencies. In line with BEL-1, the induction of wild-type *p53* expression clearly resulted in p21 activation and PARP cleavage. However, p53 protein expression and acetylation were barely detectable in RS4;11, probably due to the low percentage of transduced cells. The induction of mutant *p53* expression failed to activate p21 and hardly affected PARP cleavage (Appendix A). Flow cytometric analysis and trypan blue cell counting showed an ~8% decrease in viable RS4;11 cells and an ~8% increase in apoptosis 24 h post wild-type p53 expression induction (Appendix A).

Hence, activation of p53 in *KMT2A*-rearranged ALL cells carrying wild-type *p53* indeed leads to apoptosis induction and may represent a vulnerability to which *KMT2A*-rearranged ALL cells are susceptible.

### 3.5. MDM2 Inhibitors Are Effective against KMT2A-Rearranged ALL Cells Carrying Wild-Type P53

Drug-induced activation of p53 is most directly achieved using MDM2 inhibitors that release p53 suppression, such as Idasanutlin. We, therefore, reasoned that other MDM2 inhibitors should be equally as effective, and tested two additional MDM2 inhibitors, i.e., Milademetan (DS-3032) and AMG232, both of which are currently evaluated in clinical trials for patients diagnosed with acute myeloid leukemia (AML) and other types of cancers (e.g., NCT03671564 and NCT02016729). The dose–response curves for Milademetan and AMG232 showed that both inhibitors effectively eliminate the entire population of RS4;11 cells (carrying wild-type p53) at nanomolar concentrations (Figure 5A,B). In contrast, ALL-PO cells (carrying mutated p53) largely remained unaffected by these drugs (Figure 5A,B).

## 4. Discussion

In the present study, we performed a high-throughput drug library screening on patient-derived *KMT2A*-rearranged infant ALL cells. This led to the identification of multiple known and novel agents highly effective against *KMT2A*-rearranged infant ALL cells in vitro while only mildly affecting non-leukemic BM cells. Thus, apart from displaying strong anti-leukemic activity, such drug candidates also potentially allow for a favorable therapeutic index for bone marrow toxicity.

The dose–response curves of selected drug candidates showed that the majority of the identified agents were validated in additional primary *KMT2A*-rearranged infant ALL samples as well as in various *KMT2A*-rearranged ALL cell line models. Seven out of the eight selected drug candidates were shown to be effective in multiple cell models. This, and the fact that various agents were identified that already have been shown to be effective against *KMT2A*-rearranged acute leukemia, such as BCL-2 inhibitors [25,26], HDAC inhibitors [27,28,29], and SN-38 [21], clearly demonstrated the validity of our drug screens.

Interestingly, many of the identified drug hits were known to either directly or indirectly activate or modulate p53 signaling (Figure 1C), which was confirmed for various selected candidate drugs by western blot analysis. This suggests that drug-induced activation of p53 may represent a vulnerability to which *KMT2A*-rearranged acute leukemia cells are susceptible. In line with this, several studies reported that a variety of leukemia-driving KMT2A fusion proteins actively suppress p53 expression/activation [50,51,52]. Possibly, the manifestation of oncogenic KMT2A fusion proteins by inappropriate chromosomal translocations involving the *KMT2A* gene can only be established in the absence of a p53-mediated DNA damage response. If so, drug-induced p53 activation may well represent an essential condition for successful treatments for this aggressive type of leukemia. Obviously, apart from MDM2 inhibitors such as Idasanutlin, all the other drug classes identified here were originally designed for other purposes and inhibit a range of distinct targets that eventually and often indirectly affect p53 signaling. Hence, the anti-leukemic effects of most of these drugs rely on more than activation of p53 alone. From that perspective, MDM2 inhibition by, for instance, Idasanutlin, represents the most direct option for drug-induced activation of p53. The fact that *KMT2A*-rearranged ALL cell line models carrying *p53* mutations that most likely result in a loss of tumor suppression function (including apoptosis induction) completely fail to respond to Idasanutlin or to other MDM2 inhibitors, such as Milademetan and AMG232, underlines the specificity of these agents. At the same time, such inhibitors will almost exclusively be beneficial for patients carrying wild-type *p53*, such as *KMT2A*-rearranged infant ALL patients, where the incidence of *p53* mutations is rare and only occurs in ~4% of the cases [10,11].

Interestingly, Richmond et al. recently reported promising preclinical results with the MDM2 antagonist RG7112, a nutlin-derivative and predecessor of Idasanutlin (RG7388), in various xenograft mouse models of *KMT2A*-rearranged infant ALL [53]. Moreover, Idasanutlin in combination with chemotherapy or Venetoclax is currently progressing to phase I/II in clinical trials for relapsed/refractory pediatric leukemia patients (NCT04029688), and successful results coming forth from these trials might facilitate its implementation in the treatment of *KMT2A*-rearranged infant ALL. More importantly, Idasanutlin has been shown to be highly effective in combination with the BCL-2 inhibitor Venetoclax against *KMT2A*-rearranged acute myeloid leukemia [36]. As we identified both the BCL-2 inhibitors Venetoclax and Navitoclax as strong hits at 10 nm concentrations, we asked whether combining these agents with Idasanutlin would also be potent in *KMT2A*-rearranged ALL. Testing this in the *KMT2A*-rearranged ALL cell line model RS4;11 indeed demonstrated strong synergistic effects between Venetoclax or Navitoclax in combination with Idasanutlin (Appendix A). This demonstrates that combining different classes of the here-identified plethora of drug candidates effective against *KMT2A*-rearranged infant ALL has the potential to result in synergistic drug combinations.

Taken together, our high-throughput drug library screens provide a wealth of valid and potential drug candidates that are highly effective against *KMT2A*-rearranged infant ALL cells while showing far less toxicity towards non-leukemic BM, encouraging further preclinical testing. In addition, our findings collectively suggest that *KMT2A*-rearranged infant ALL cells may well be vulnerable to (drug-induced) activation of wild-type p53, which is in concordance with earlier reported literature.

## Figures and Tables

**Figure 1 biomedicines-10-00638-f001:**
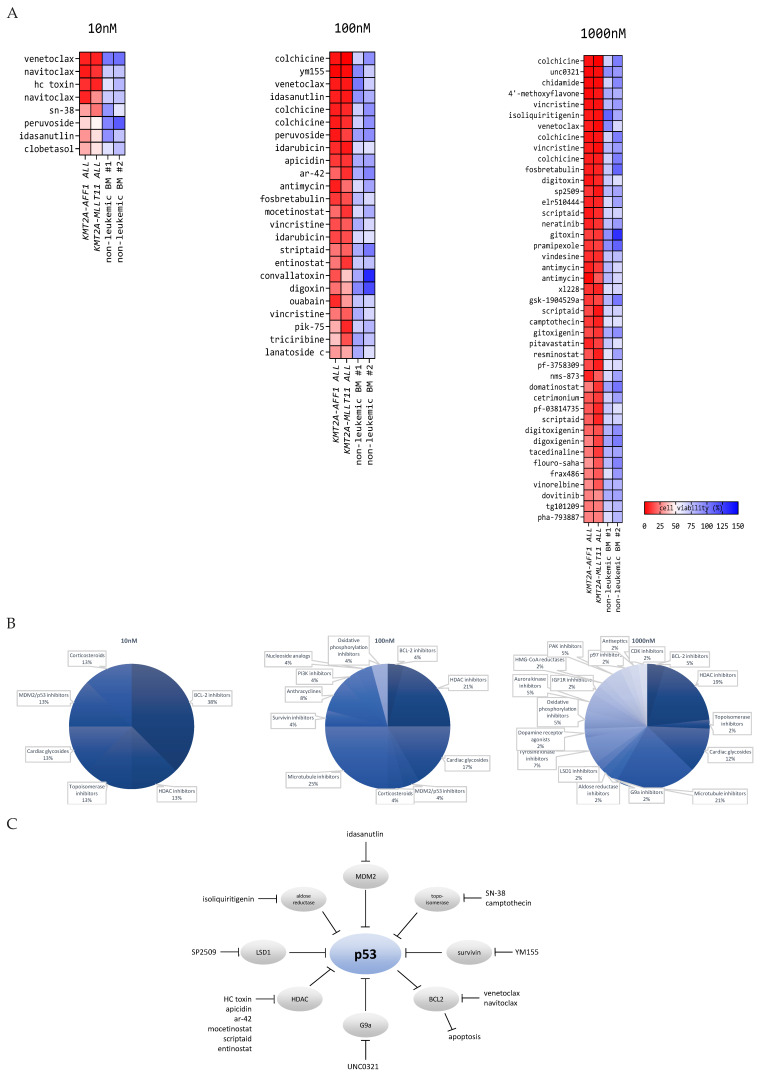
Drug candidates identified in the high-throughput screening on primary *KMT2A*-rearranged infant ALL cells and non-leukemic bone marrow cells favor p53 signaling: (**A**) heatmap representation of the in vitro cytotoxicity in primary *KMT2A*-rearranged infant ALL cells and non-leukemic bone marrow cells exposed to 10, 100, and 1000 nm drug. Drugs shown in the heatmap are drug candidates identified in the high-throughput screening. Drugs are ranked based on the biggest difference between the average cell viability of the leukemic samples versus the average cell viability of the non-leukemic bone marrow samples. Drug hits occurring multiple times in the heatmap indicate that the drug was present in more than one of the tested drug libraries. (**B**) Pie charts of the most recurrent drug classes and targets identified by high-throughput drug screening. (**C**) Schematic overview of drug classes and targets related to p53 signaling.

**Figure 2 biomedicines-10-00638-f002:**
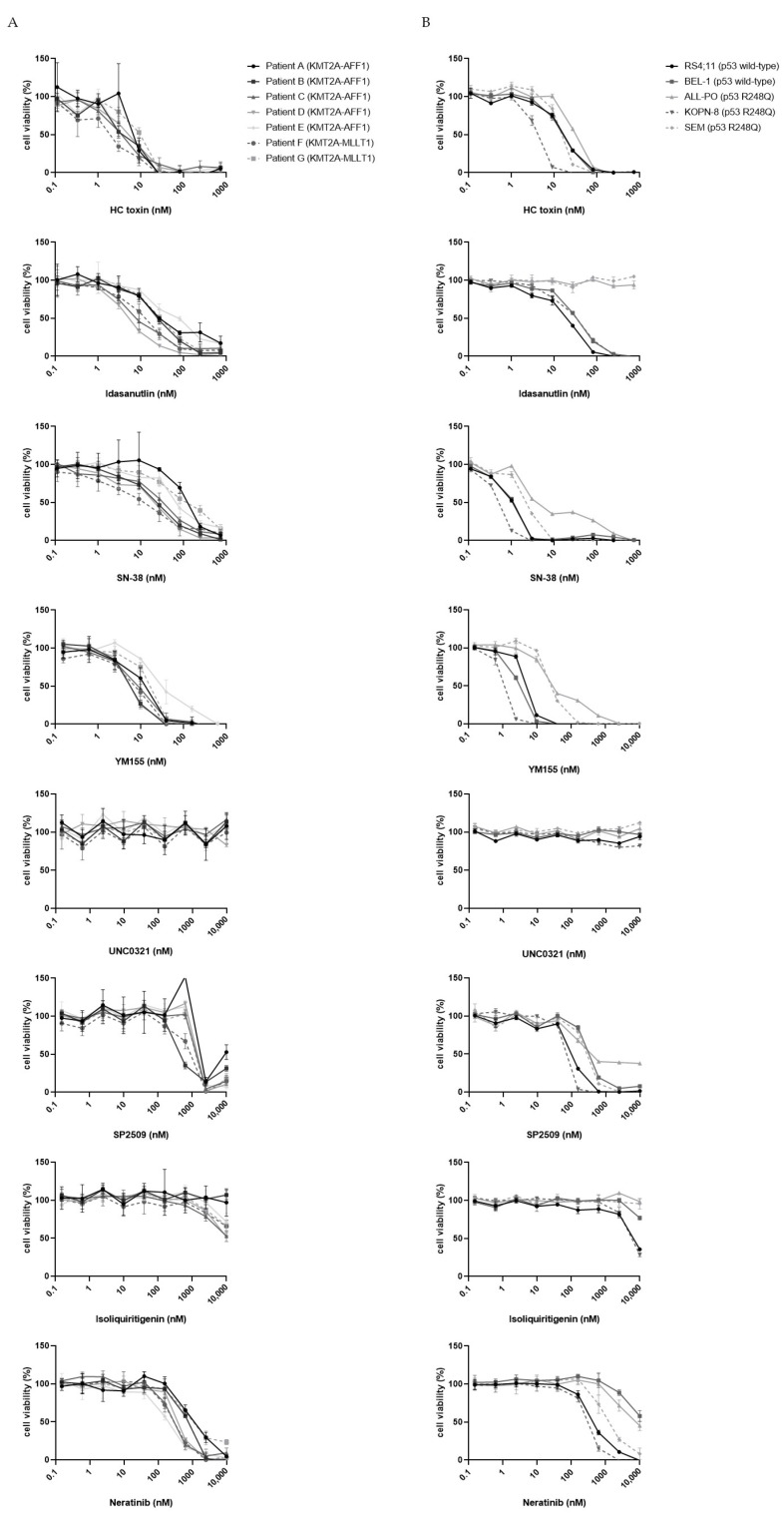
Validation of candidate drugs on patient-derived *KMT2A*-rearranged infant ALL cells and *KMT2A*-rearranged ALL cell lines: broad-range dose–response curves of eight candidate drugs on (**A**) primary *KMT2A*-rearranged infant ALL patient samples (*n* = 7) and (**B**) various *KMT2A*-rearranged ALL cell line models (*n* = 5). Cell viability is based on a 4-day MTT assay and normalized to DMSO controls. Data points are represented as the mean ± SD of *n* = 3 technical replicates.

**Figure 3 biomedicines-10-00638-f003:**
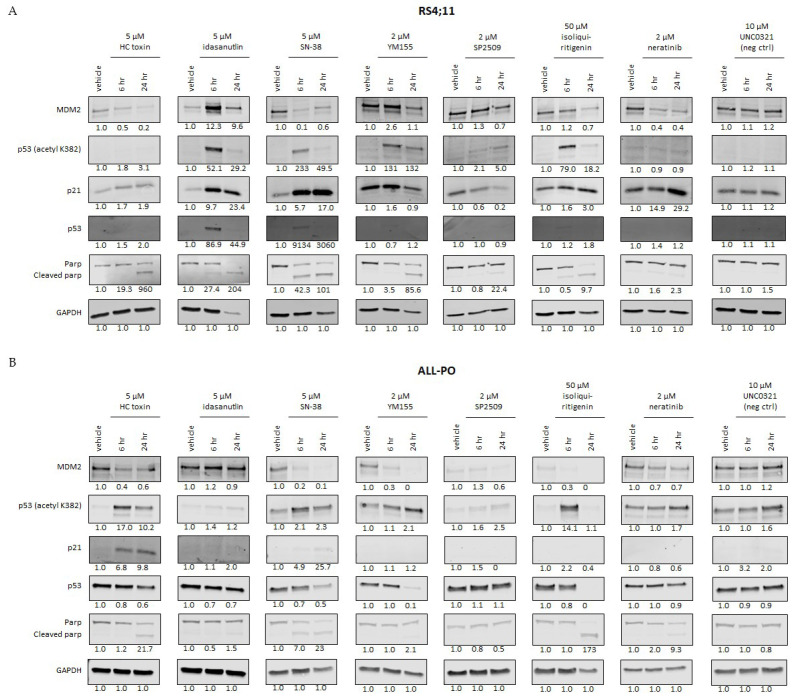
Candidate drugs affect the expression of the p53 pathway in *KMT2A–AFF1* ALL cells: western blot analysis showing the protein expression of MDM2, p53 acetylation at lysine 382, p21, p53, and cleaved PARP upon exposure to eight candidate drugs at indicated concentrations or vehicle controls for 6 and 24 h in the cell lines (**A**) RS4;11 and (**B**) ALL-PO. Expression of GAPDH was used as a loading control. Quantification of the expression was relative to GAPDH and normalized to vehicle controls. The ratio cleaved PARP:PARP was used for quantification of PARP cleavage. UNC0321 was used as negative control.

**Figure 4 biomedicines-10-00638-f004:**
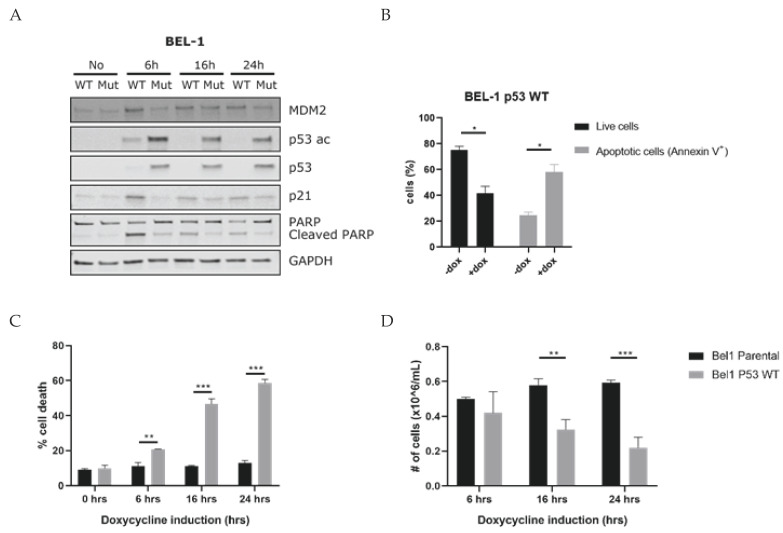
Forced p53 activation induces apoptosis in BEL-1 cells carrying wild-type *p53*: (**A**) western blot analysis showing the protein expression of MDM2, p53 acetylation at lysine 382, p53, p21, and cleaved PARP measured at 6, 16, and 24 h post doxycycline induction in the transduced BEL-1 cell line carrying doxycycline-inducible expression vectors either encoding wild-type or mutated (R248Q) *p53*. Housekeeping gene GAPDH was used as loading control. (**B**) Percentage of viable and apoptotic cells in BEL-1 cells carrying no vector (parental) or wild-type *p53*, at 24 h post p53 induction by doxycycline determined by flow cytometric analysis of the Annexin V/7AAD staining. (**C**) Percentage of dead cells and (**D**) percentage of viable cells, measured by a manual cell count of Trypan Blue positive cells BEL-1 cells and Trypan Blue negative BEL-1 cells carrying either no vector (parental) or wild-type *p53*, measured at 6, 16 and 24 h post p53 induction by doxycycline. All data points are represented as the mean ± SEM of *n* = 3. Cell viability comparisons between *KMT2A*-rearranged ALL cell line conditions were performed using the multiple *t*-test of average; * = *p* ≤ 0.05; ** = *p* ≤ 0.01; *** = *p* ≤ 0.001.

**Figure 5 biomedicines-10-00638-f005:**
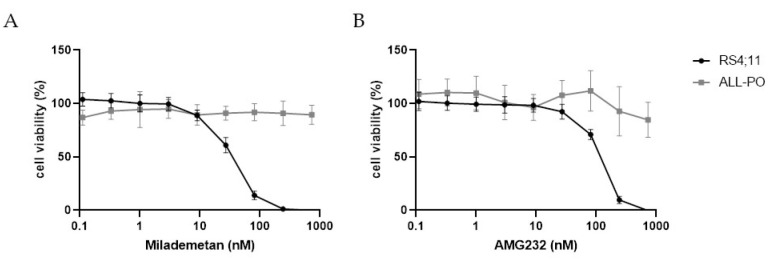
Efficacy of MDM2 inhibitors in *KMT2A*-rearranged ALL cells carrying wild-type *p53* or mutated *p53*: in vitro drug responses of (**A**) Milademetan (DS-3032) and (**B**) AMG232 on KMT2A–AFF1 cell lines RS4;11 (*p53* wild-type) and ALL-PO (mutated *p53*) determined by a 4-day MTT assay. Cell viability was normalized to DMSO controls. Data points are represented as the mean ± SD of *n* = 7 technical replicates.

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
