# Peer review of "High-Throughput Drug Library Screening in Primary *KMT2A*-Rearranged Infant ALL Cells Favors the Identification of Drug Candidates That Activate P53 Signaling"

_biomedicines, 2022, doi:10.3390/biomedicines10030638_

Round 1

Reviewer 1 Report

In this study, Wander et al. have performed an extensive (4191 compounds), high-throughput screen of re-purposed and mostly FDA-approved drugs to find novel agents that might be effective against highly aggressive KMT2A-rearranged infant ALL. Importantly, the screen was performed on primary patient samples, and positive hits that seemed to leave normal bone marrow cells unaffected were confirmed on additional patient samples and a range of leukaemia cell lines. Interestingly, many of the compounds that have shown an effect directly or indirectly target the p53 pathway. And since the p53 pathway as a potential therapeutic target in infant MLL-r leukaemia has only fairly recently started to be explored (ref. 52) and many compounds are available, the authors decided to concentrate on the p53 pathway in subsequent experiments. Specifically, they test the effect of the selected drugs on p53 expression, p53 acetylation, p21 expression, MDM2 expression and Parp cleavage in two different leukaemia cell lines (one of which carries a p53 mutation), as well as the effects of forced p53 expression and two additional MDM2 inhibitors on leukaemia cell line survival.

Overall, this is a scientifically sound study that will be of interest to the field, and I only have a few queries as specified below.

Specific comments

  1. The authors claim on several occasions throughout the manuscript that MLL-r infant ALL cells are “particularly” vulnerable to p53 activation (p.1, line 37; p.2, line 43; p.11, line 12; p.12, line 29; p.13, line 23), but do not really show any evidence for that. To substantiate this claim, they would also have to test other, non-MLL-r leukaemia cell lines for susceptibility to the drugs and to activation of the p53 pathway. And with this being such a central pathway in cancer biology and in protecting cells from malignant transformation, would one not expect most cells to undergo apoptosis when p53 is overexpressed?
  2. The writing in Figure 1B is very small and difficult to read. It would be very helpful if the font could be increased.
  3. The authors should provide more information on how the R248Q mutation affects p53 function. On p. 9, line 30, they state that it “abrogates most of the cellular responses mediated by wild-type p53 and has gain-of-function activity”. What are these gains-of-function? There does not seem to be any effect on MDM and p21 expression or Parp cleavage in Fig. 4A. And would this mutation explain all of the different drug responses of RS4;11 cells compared with ALL-PO cells in Fig. 3?
  4. On p.10, lines 14-17, the authors suggest that induction of wild-type p53 resulted in an increase in p53 protein expression, as shown in Fig. 4A, but there is in fact very little p53 expression. Would higher levels of p53 not be toxic to any cell?
  5. On p.9 and 10, the authors mention a negative feedback mechanism triggered by p53 activation and ultimately leading to its own degradation. What is the evidence that this is really what causes the waning of the effect at 16h and 24h, e.g. in Fig. 4A? It would also be helpful if the authors could provide more information on this feedback mechanism and cite the relevant literature.
  6. In Fig. 5, the authors test two additional MDM2 inhibitors, Milademetan (DS-3032) and AMG232. Were these not included in the 4191 compounds of the initial screen?

Reviewer 2 Report

This paper describes the results of a drug screen testing FDA-approved for their effect on KMT2A rearranged ALL. The authors found that many of the positive hits from their screen activate p53, leading to the conclusion that KMT2A rearranged ALL is sensitive to p53 induced apoptosis. However, this conclusion is in large part based on western blots lacking any statistical analysis or indication of the number of experiments performed, and are in some cases too light to be able to determine if they support the authors' conclusions. 

Minor concerns:

-The text in Fig. 1B is too small to read.

-The addition of mutational status for each cell line to the figure legend in 2B would be helpful. 

Round 2

Reviewer 1 Report

The authors have addressed all of my comments and I have no further queries.

Reviewer 2 Report

The authors have addressed my concerns satisfactorily.